# OpenReview forum: "STaR-Attack: A Spatio-Temporal and Narrative Reasoning Attack Framework for Unified Multimodal Understanding and Generation Models"
_ICLR.cc/2026/Conference — ICLR 2026 Conference Withdrawn Submission_

### Official Review · Reviewer_2hrQ · 2025-10-26

**Soundness:** 2
**Presentation:** 3
**Contribution:** 2
**Rating:** 4
**Confidence:** 3

**Summary:**

This paper proposes STaR-Attack, a multi-turn spatio-temporal and narrative reasoning jailbreak targeting unified multimodal models (UMMs). It exploits a newly defined Cross-Modal Generative Injection (CMGI) vulnerability by crafting a three-act story (setup -> hidden climax -> resolution) that embeds malicious instructions into model-generated images, which are later interpreted by the understanding module. The attack integrates a guess-and-answer mechanism and dynamic difficulty adjustment to maintain semantic coherence and improve attack success. Experiments on AdvBench and HarmBench demonstrate high attack success rates and improved relevance over recent baselines like FlipAttack and ReNeLLM.

**Strengths:**

1. Novel integration of narrative reasoning and dynamic difficulty into multimodal jailbreaks.

2. Clear identification of generation–understanding coupling (CMGI) as a new vulnerability surface.

3. Strong quantitative gains on standard safety benchmarks with detailed ablations.

**Weaknesses:**

1. Realism gap: assumes access to unrestricted generation modules, limiting real-world applicability.

2. Incremental novelty: similar attack ideas (multi-turn, visual priming, contextual prompting) exist - as described in related work section (" Visual Contextual Attack Miao et al. (2025a) demonstrates that injecting or synthesizing image cues aligned
with textual prompts substantially amplifies attack success, while Response Attack (Miao et al.,2025b)"

3. Missing reference; recent visual adversarial works (e.g., On the robustness of large multimodal models against image adversarial attacks CVPR'24, X Cui).

4. Limitation on attack applicability analysis - need to test on stronger models; testing on 7B scale models and Gemini-2.5 flash shows limitations on deeper investigation when testing this attacks on much stronger models e.g., Gemini-2.5 Pro or GPT-5.

**Questions:**

1. How would STaR-Attack perform under realistic API restrictions (separate generation/understanding safety layers)?

2. What is the query efficiency and computational cost compared to black-box baselines like PAIR or universal image attacks?

3. Test results on stronger models (Gemini 2.5 Pro, GPT-5)

---

### Official Review · Reviewer_UMHz · 2025-10-31

**Soundness:** 2
**Presentation:** 3
**Contribution:** 2
**Rating:** 4
**Confidence:** 3

**Summary:**

This paper exploits a security vulnerability, Cross-Modal Generative Injection (CMGI), which is inherent to unified multimodal understanding and generation Models (UMMs).
The authors believe that the vulnerability stems from the tight coupling of generation and understanding capabilities within a single model.
An attacker can leverage this coupling by first using the model's generative function to create an information-rich adversarial image and then forcing the model's understanding function to absorb this malicious content in a single, coherent process. This allows for the injection of a large amount of harmful information that can bypass the model's safety alignments.
To systematically exploit this vulnerability, the authors propose STaR-Attack, a multi-turn jailbreak framework. The attack is built on a spatio-temporal and narrative reasoning structure. It conceals a malicious event (the "climax") within a three-act narrative, generating only the pre-event ("setup") and post-event ("resolution") scenes using the UMM's own image generation capability. The attack is then executed through a "guess and answer" game, where the original malicious query is embedded among benign candidates. The model is prompted to select and answer the most relevant question based on the narrative context it helped create, thereby recovering and responding to the harmful query without any semantic drift.
A dynamic difficulty mechanism is also introduced, which adjusts the number of benign candidates based on the model's responses to improve attack success and stability.

**Strengths:**

+ clear presentation
+ sound design
+ comprehensive evaluation

**Weaknesses:**

1. The paper's central claim is the identification of a fundamental vulnerability in UMMs called Cross-Modal Generative Injection. However, the evidence provided primarily demonstrates the effectiveness of the highly complex, multi-turn STaR-Attack framework itself. It remains unclear whether the success is due to the fundamental CMGI vulnerability or the sophisticated, specifically engineered narrative and game mechanics.

2. The dynamic difficulty mechanism is shown to improve ASR, but the explanation for why it works is largely hypothetical. The paper lacks analysis to validate these hypotheses, such as how the model's internal attention or reasoning processes change with increasing difficulty.

3. The paper ignores the practical cost of the attack. The paper should discuss the attack's computational and temporal cost (e.g., average number of API calls, total inference time). An discussion of this cost help clarify the real-world threat levels.

4. While the paper demonstrates attack capability, it dose not test or discuss any concrete mitigation strategies.

**Questions:**

1. It is suggested to better clarify the reason why attack success.

2. It is suggested to show more solid evidence to explain why dynamic difficulty mechanism improve ASR.

3. It is suggested to analyze the actual cost of attack.

4. It is suggested to discuss potential mitigation strategies.

---

### Official Review · Reviewer_8jvv · 2025-11-03

**Soundness:** 3
**Presentation:** 3
**Contribution:** 2
**Rating:** 4
**Confidence:** 4

**Summary:**

This paper focuses on Unified Multimodal Understanding and Generation Models (UMMs), identifies the Cross-Modal Generative Injection (CMGI) vulnerability, and proposes STaR-Attack (Spatio-Temporal and Narrative Reasoning Attack) to address the shortcomings of existing attacks—such as being limited to a single modality and prone to semantic drift. With the three-act narrative structure as its core, STaR-Attack achieves jailbreak by utilizing the capabilities of UMMs through multi-turn interactions.

**Strengths:**

1. This paper proposes a new attack framework that demonstrates excellent attack effectiveness with a very high success rate.
2. The writing structure of the paper is clear and the illustrations are easy to read.
3. Conducted ablation experiments to verify the effectiveness of each part of the method, and the experiments were relatively complete.

**Weaknesses:**

1. Although the results show significant improvements with this method, the evaluated models have relatively weak security defenses. The open-source models used only include the 7B version, and the closed-source models are limited to the Gemini Flash series. It would be beneficial to include larger models with more parameters, such as 13B-70B, as well as other models like Gemini-Pro, GPT-4o, and GPT-5.
2. The paper only provides an attack framework, and although the results show significant improvements and ablation of each component, it still focuses primarily on jailbreak success rates, lacking deeper insights and interpretability analysis. The exploration of the essence of the CMGI vulnerability is insufficient. For example, analyzing how the model allocates attention to different modalities or malicious queries during inference and providing a defense strategy specifically targeting this phenomenon would be valuable.

**Questions:**

1. How does the attack perform on stronger models？
2. Can you provide a more fundamental analysis rather than a jailbreak success rate to explain the significant improvement shown in the results？

---

### Official Review · Reviewer_g9qP · 2025-11-05

**Soundness:** 2
**Presentation:** 2
**Contribution:** 2
**Rating:** 2
**Confidence:** 4

**Summary:**

The paper uncovers a Cross-Modal Generative Injection (CMGI) vulnerability in unified multimodal models and proposes STaR-Attack, a three-act spatio-temporal multi-turn jailbreak that hides a malicious climax between generated setup images and uses dynamic-difficulty Q&A to achieve high attack success on benchmarks.

**Strengths:**

The evaluation is tested on AdvBench and HarmBench across open- and closed-source UMMs (e.g., BAGEL, Janus-Pro, Gemini), reporting both ASR and the relevance-aware RASR

**Weaknesses:**

1.	I believe there is an inherent contradiction in the attack setting: the paper assumes the attacker already possesses a Malicious LLM that can output any harmful content they desire. However, instead of directly using this capability, the authors go through multiple layers of wrapping and transformation to make another LLM generate harmful content. In my view, these subsequent steps are redundant, because the attacker’s goal is already achieved at the first stage (Malicious LLM). In reality, most ordinary users do not have access to such a Malicious LLM, and jailbreaking under this constraint would be closer to real-world scenarios.
2.	The number of models tested and baselines compared in this paper is quite limited. For instance, GPT, Qwen, and Deepseek were not evaluated, and many well-known jailbreaking methods such as AutoDAN, AutoDAN-reasoning, GCG, PAIR, AutoAdv, ArtPrompt, etc., were not included for comparison. This lack of breadth undermines the persuasiveness of the results.

**Questions:**

See the weakness part of my review

---

### Official Review · Reviewer_mcRB · 2025-11-06

**Soundness:** 3
**Presentation:** 3
**Contribution:** 2
**Rating:** 4
**Confidence:** 5

**Summary:**

This paper introduces STaR Attack, a multi turn jailbreak framework that targets unified multimodal models by exploiting the coupling between generation and understanding. The method hides a malicious event inside a three act narrative, uses the model to generate setup and resolution images, then runs a guess and answer round that selects the original harmful query from benign candidates. A dynamic difficulty schedule adapts the candidate set size to improve stability. Experiments on BAGEL, Janus Pro, and Gemini Flash show high attack success and stronger relevance to the original query than prior methods, suggesting a real CMGI vulnerability in UMMs.

**Strengths:**

Clear articulation of a new threat surface in UMMs through cross modal generative injection.
Elegant attack design that avoids semantic drift via narrative framing plus a candidate selection game.
Dynamic difficulty improves reliability, with solid ablations on interaction structure and difficulty distribution.
Evaluation spans open and closed models, reports both ASR and a relevance aware metric, and analyzes judge effects.
Implementation details and prompts in the appendix improve transparency and reproducibility.

**Weaknesses:**

Dependence on an uncensored external model to craft scene descriptions may inflate feasibility and limits the closed loop threat model.
Heavy use of proprietary judges and embeddings could bias ASR and relevance outcomes, with limited human evaluation.
Assumes the platform permits image generation of risky context in the early rounds, which many deployments filter or disable.
Limited analysis of cost, latency, and rate limiting in multi turn settings, and no study of long context truncation effects.
Defense discussion is brief, and sensitivity to key hyperparameters such as similarity thresholds and schedule choices is under explored.

**Questions:**

What is the concrete deployment threat model where an attacker can reliably access both generation and understanding in one session across turns?
How does the method fare when image generation is safety filtered or outsourced to a separately aligned service that sanitizes content?
How sensitive is success to the narrative template, the difficulty schedule, and the specific embedding model used for relevance?

---

### Note · Authors · 2025-11-14

I have read and agree with the venue's withdrawal policy on behalf of myself and my co-authors.